# Structure of O-Polysaccharide and Lipid A of *Pantoea Agglomerans* 8488

**DOI:** 10.3390/biom10050804

**Published:** 2020-05-22

**Authors:** Tetiana V. Bulyhina, Evelina L. Zdorovenko, Ludmila D. Varbanets, Alexander S. Shashkov, Alexandra A. Kadykova, Yuriy A. Knirel, Oleh V. Lushchak

**Affiliations:** 1D.K. Zabolotny Institute of Microbiology and Virology (IMV), The National Academy of Sciences, 154 Zabolotnoho Str., 03143 Kyiv, Ukraine; 2N.D. Zelinsky Institute of Organic Chemistry, Russian Academy of Sciences, Moscow 119991, Russia; zdorovenkoe@mail.ru (E.L.Z.); varbanets_imv@ukr.net (L.D.V.); shash@ioc.ac.ru (A.S.S.); yknirel@gmail.com (Y.A.K.); 3Higher Chemical College of the Russian Academy of Sciences, D.I. Mendeleev University of Chemical Technology of Russia, Moscow 125047, Russia; alexandra-kaa@mail.ru; 4Department of Biochemistry and Biotechnology, Natural Sciences Institute, Vasyl Stefanyk Precarpathian National University, 76018 Ivano-Frankivsk, Ukraine; oleh.lushchak@pnu.edu.ua

**Keywords:** *Pantoea agglomerans*, lipopolysaccharide, O-polysaccharide structure, lipid A, toxicity, pyrogenicity

## Abstract

The *Pantoea agglomerans* 8488 lipopolysaccharide (LPS) was isolated, purified and characterized by monosaccharide and fatty acid analysis. The O-polysaccharide and lipid A components of the LPS were separated by mild acid degradation. Lipid A was studied by electrospray ionization mass spectrometry (ESI-MS) and found to consist of hexa-, penta-, tetra- and tri-acylated species. Two-dimensional nuclear magnetic resonance (NMR) spectroscopy revealed the following structure of the O-polysaccharide repeating unit →3)-α-L-Rha*p*-(1→6)-α-D-Man*p*-(1→3)-α-L-Fuc*p*-(1→3)-β-D-GlcNAc*p*-(1→. The LPS showed a low level of toxicity, was not pyrogenic, and reduced the adhesiveness index of microorganisms to 2.12, which was twofold less than the control. LPS modified by complex compounds of germanium (IV) and tin (IV) were obtained. It was found that six LPS samples modified by Sn compounds and two LPS samples modified by Ge compounds lost their toxic activity when administered to mice in a dose of LD_50_ (105 µg/mice or 5 mg/kg). However, none of the modified LPS samples changed their serological activity in an Ouchterlony double immunodiffusion test in agar.

## 1. Introduction

*Pantoea agglomerans* (formerly *Enterobacter agglomerans*, *Erwinia herbicola*) is a ubiquitous bacterial strain that belongs to the *Enterobacteriaceae* family [1]. It is primarily an environmental and agricultural organism that inhabits plants, soil, water, air and dust. However, *P. agglomerans* strains are frequently found in humans and isolated in hospitals. They might affect various organs in the body, ranging from non-life-threatening skin infections to fatal multiorgan systemic disorders [2,3]. However, *P. agglomerans* strains can also have a positive effect by stimulating the production of substances effective for the treatment of cancer and other diseases [3,4]. Bacterial lipopolysaccharide (LPS), a major component of the outer membrane of Gram-negative bacteria, is involved in both these positive and negative processes [5]. LPS also is known as a potent immunostimulator in both humans and animals.

The structural basis of *P. agglomerans* LPS is similar to other Gram-negative species, but some distinctive features have also been reported. Tsukioka et al. [6] determined the structure of lipid A of this preparation and found that *P. agglomerans* LPS comprised at least two types of lipid A with different levels of acylation. These lipid A variants resembled the hexaacyl lipid A of *Escherichia coli* and heptaacyl lipid A of *Salmonella minnesota*. Although the biological activity of *P. agglomerans* LPS is mainly related to the structure of lipid A, some authors also attribute an important role to the structure of the O-specific polysaccharide (OPS) as a determinant of this activity. Karamanos et al. [7] found that the OPS consist of repetitive pentasaccharide units containing *N*-acetyl-L-Fucosamine (FucNAC), *N*-acetyl-D-Glucosamine (GlcNAc), Rhamnose (Rha) and Glucose (Glc). Hashimoto et al. [8] established that the OPS studied was composed of linear tetrasaccharide repeating units, comprised of glucose and rhamnose, where 40% of one of the rhamnose residues was substituted by glucose. Cimmino et al. [9] determined the structure of the OPS of *P. agglomerans* LPS-PW which consisted of linear tetrasaccharide repeating units containing D-rhamnose residues [10]. Despite extensive literature on the biological activity of *P. agglomerans*, investigations of LPS phytopathogenesis are limited.

Therefore, this work is dedicated to the isolation and characterization of LPS from *P. agglomerans* 8488 isolated from oat [6,11,12,13,14]. The structural and functional peculiarities of LPS are used as one of the recognized chemotaxonomic criteria in the systematics of Gram-negative bacteria. Moreover, subtle variations in LPS structures are the molecular basis for the development of intraspecies serological classification schemes. Until now, for systematics a heterogeneous species *P. agglomerans* have many unresolved issues. The goal of this work was to isolate, chemically characterize, and study the functional and biological activities of the *P. agglomerans* 8488 LPS, as well as elucidate the peculiarity of OPS and lipid A structure of this strain.

## 2. Materials and Methods

### 2.1. Growth of Bacteria, Isolation and Degradation of the LPS

*P. agglomerans* strain 8488 taken from the live culture collection of the Department of Phytopathogenic Bacteria of IMV (isolated from oat) was grown on a potato agar medium (500 g/L potato, 15 g/L agar and 5 g/L NaCl) at 28–30 °C for 36 h. Cells were collected at the end of the logarithmic phase of growth with physiological solution (0.9% solution of NaCl) and separated by centrifugation (20 min, 5000× *g*). The precipitated cells were washed twice with saline solution (0.9% NaCl) and dried with acetone and diethyl ether. The lipopolysaccharides were isolated by phenol-water extraction [15], followed by the removal of nucleic acids by precipitation with aqueous 50% trichloroacetic acid and ultracentrifugation at 105,000× *g* (4 h, 3 times).

### 2.2. Assay of Carbohydrates, Nucleic Acids and Proteins

The amount of carbohydrates was determined by the Dubois method [16]. The content of the carbohydrates was determined in accordance with the glucose standard calibration curves. The nucleic acids and proteins were analyzed by the Spirin [17] and Lowry [18] methods, respectively. A method for nucleic acids identification was based on spectrophotometric determination. The 2-keto-3-deoxyoctonic acid (KDO) was quantified by reaction with thiobarbituric acid (Osborn method) [19].

### 2.3. Fatty Acid Analysis

The fatty acid composition was determined after the hydrolysis of LPS in the presence of 1.5% acetyl chloride in methanol (100 °C, 4 h). The fatty acid methyl esters were analyzed by an Inert chromatography-mass spectrometric system 6890N/5973 (Agilent, CA, USA) equipped with an HP-5ms column using the temperature program from 150 to 250 °C at 4 °C/min; helium was used as the carrier gas at the flow rate of 1.2 mL/min. The fatty acids were identified using the standard mixture of the fatty acid methyl esters and the available Personal Computer database [20]. The quantitative ratios of individual fatty acids were expressed as a percentage of the total sum of peak areas.

### 2.4. Analyses of Monosaccharides

The identification of the neutral monosaccharides was carried out after the hydrolysis of the LPS preparations in 2 M HCl (5 h, 100 °C). Monosaccharides were analyzed as the alditol acetates [21] by Inert chromatography-mass spectrometry system 6890N/5973 (Agilent, Santa Clara, CA, USA) equipped with a DB-225mS column. The carrier gas was helium at a flow rate of 1 mL/min. The temperatures of the evaporator, interface and thermostat were 250, 280, and 220 °C, respectively (isothermal mode). The monosaccharides were identified by comparing the retention times of polyol acetates in the experimental and standard samples and identified using the ChemStation (Agilent Technologies, San Diego, CA, USA) database. The monosaccharide composition of LPS was expressed in percentage of the total sum of peak areas.

### 2.5. Isolation of the O-Polysaccharide

An LPS sample from *P. agglomerans* 8488 was hydrolyzed in aqueous 2% acetic acid (AcOH) at 100 °C for 1.5 h. The lipid precipitate was removed by centrifugation (13,000× *g*, 20 min), and the carbohydrate portion was fractionated by gel-permeation chromatography on a Sephadex G-50 Superfine column (56 × 2.6 cm) (Amersham Biosciences, Uppsala, Sweden) in a 0.05 M pyridinium acetate buffer (pH 4.5) monitored with a differential refractometer (Knauer, Berlin, Germany) to give an O-polysaccharide fraction.

### 2.6. Sugar Analysis

An O-polysaccharide sample (0.5 mg) was hydrolyzed in 2 M CF_3_CO_2_H (120 °C, 2 h). Monosaccharides were analyzed as the alditol acetates by gas-liquid chromatography (GLC) on a HP-5 capillary column using the Maestro (Agilent 7820) chromatograph (Interlab, Moscow, Russia) and a temperature gradient of 160 °C (1 min) to 290 °C at 7 °C/min [20]. The absolute configuration of mannose was determined by the GLC of the acetylated (*S*)-2-octyl glycosides as described by the authors of [22].

### 2.7. Nuclear Magnetic Resonance (NMR) Spectroscopy

Samples were deuterium-exchanged by freeze-drying twice from D_2_O and then examined as a solution in 99.9% D_2_O. The ^1^H- and ^13^C-NMR spectra were recorded using a Bruker Avance II 600 MHz spectrometer (Karlsruhe, Germany) at 45 °C using internal sodium 3-trimethylsilylpropanoate-2,2,3,3-d4 (δ_H_ 0.0 ppm) as a reference. Two-dimensional experiments were performed using standard Bruker software, and the Bruker TopSpin 2.1 program was used to acquire and process the NMR data. A mixing time of 100 ms was used in TOCSY (Total Correlation Spectroscopy) and ROESY (Rotating-frame Overhauser Spectroscopy) experiments. The ^1^H, ^13^C HMBC (Heteronuclear Multiple Bond Correlation experiment) spectra were recorded with a 60-ms delay for the evolution of long-range couplings.

### 2.8. Mass Spectrometry

Negative ion mode high-resolution electrospray ionization (ESI) mass spectrum was measured on a Bruker micrOTOF II mass spectrometer (Münster, Germany) [23]. The interface capillary voltage was 3200 V, the mass ranging from *m/z* 50 to 3000 Da. A syringe injection was used for 1:1:0.1 acetonitrile/water/Et3N solutions at a flow rate of 3 µL/min. Nitrogen was applied as dry gas; interface temperature was set at 180 °C. Internal calibration was done with Electrospray Calibrant Solution (Fluka, Heidelberg, Germany).

### 2.9. Determination of the Sensitivity of the Microbial Culture to Polymyxin B

The sensitivity of *P. agglomerans* cultures to polymyxin B was determined by the disk-diffusion agar method. The diameter (d) of growth inhibition of bacterial cultures around the disk with the antibiotic was measured with an accuracy of 1 mm. Strain sensitivity was determined in accordance with established NCCLS (National Committee for Clinical Laboratory Standards) standards.

### 2.10. Methods of LPS Chemical Modification

Chemical detoxication allows us to obtain derivatives that retain all the positive properties of LPS (adjuvant, induction of cytokine synthesis, activation of macrophages, etc.) while being devoid of pyrogenicity and toxicity. Complexes of tin (IV) and germanium (IV) with different ligands (Table 1) were used for LPS chemical modification. Working solutions of complex compounds were prepared by dilution in dimethyl sulfoxide (DMSO) at a final concentration of 0.02 mM. LPS were dissolved in 75% DMSO at room temperature. The modification was carried out by discharging certain volumes of the LPS samples and complexes to maintain a ν(LPS)/ν(complex compounds) ratio of 1:20. The resulting solutions were sealed and kept for 1–1.5 h at 80 °C until dissolving was complete. Final volumes were adjusted to 2 mL with DMSO.

The initial complexes were synthesized at the Department of General Chemistry and Polymers of Odessa I.I. Mechnikov National University by the interaction of R-benzoylhydrazones (R = H, 2–OH, 2–OCH_3_, 2–NH_2_), R,R′-benzoylhydrazones (R = 3–Br, R′ = 5–OH), nicotinoyl-, isonicotinoylhydrazones 4–N(CH_3_) 2 benzoic (R–HBdb, 3–Br–5–OH–HBdb, HNdb, HIdb, respectively) and 2–OH-benzoic aldehydes (R–H_2_Bs, 3–Br–5–OH–H_2_Bs, H_2_Ns, H_2_Is, respectively) with tin and germanium tetrachlorides, and were comprehensively characterized by a combination of modern research methods, in particular X-ray structural analysis.

### 2.11. Determination of LPS Toxicity

The determination of LPS toxicity was carried out in healthy white non-pedigreed mice of both sexes weighing 19–21 g, which were not used previously for any experiments, sensitized with galactosamine. For this purpose, 0.5 mL of 3.2% D-galactosamine hydrochloride solution in apyrogenic sterile 0.9 %NaCl solution (saline) was injected intraperitoneally. Immediately after that, 0.2 mL of LPS in saline heated to 37 °C was injected intraperitoneally at a rate of 0.1 mL/s. In a series of LPS dilutions, the preparation dose causing the death of 50% tested animals (LD_50_) was determined and its value was used for the assessment of the LPS toxicity. Ten mice were tested is each group. The control mice were injected with 0.2 mL of sterile saline together with d-galactosamine hydrochloride. The animals were observed for 48 h [24]. The lethal dose LD_50_ was calculated by the Spearman–Karber method [25]. The work was conducted in accordance with the “General Ethical Principles of Animal Experiments”.

### 2.12. Determination of LPS Pyrogenicity

LPS pyrogenicity was determined in rabbits (chinchilla breed; age, 1–1.5 years; weight, 2.0–3.5 kg) [26]. Thermometry was performed with an electronic thermometer (Omron Matsusaka Co. Ltd., Matsusaka, Japan), which was inserted into the rectum to a depth of 5–7 cm (depending on the weight of the rabbit). All rabbits were pretested for immunoreactivity by intravenous injection of 10 mL/kg of 0.9% sterile saline. The tested LPS preparation was dissolved in a sterile saline, incubated for 10 min at 37 °C before the injection, and introduced intravenously (1 mL/kg of animal weight). The minimal pyrogenic dose of the LPS preparation was determined in a series of dilutions from 0.50 to 0.01 mg/mL and was 7.5 × 10^−3^ µg/mL. Each series of the solutions was tested in three rabbits of similar weight (the difference did not exceed 0.5 kg).

Prior to the injection of the LPS solution, the temperature of the rabbits was measured twice within a 30-min interval. The result of the last measurement was taken as the initial temperature. The LPS solution was injected no later than 15–20 min after the last temperature measurement. After the injection, the measurements were made three times at 1-h intervals. The tested LPS solution was considered apyrogenic if the total body temperature increases during 3 h did not exceed 1.4 °C. The work was conducted in accordance with the “General Ethical Principles of Animal Experiments”.

### 2.13. Serological Studies

For O-antiserum preparation, heated cells of *P. agglomerans* were used (2.5 h, boiling water bath). The cell concentration was 2 × 10^9^/mL. The rabbits were immunized intravenously five times with 4-day intervals (from 0.1 to 1.0 mL). Blood was collected on the seventh day after the last immunization. The antigenic activity of LPS was studied using the Ouchterlony double immunodiffusion method in agar [27]. The LPS isolates from 14 *P. agglomerans* strains (from different plants and geographic areas) were used as antigens.

### 2.14. Participation of the LPS in the Adhesion Processes

The participation of the LPS in the adhesion process was investigated at different concentrations of *P. agglomerans* LPS (from 0.012 to 3 mg/mL) on the adhesion of *E. coli* ATCC 25922 cells (reference-strain for the control of bacterial sensitivity to antibiotics) to native rabbit erythrocytes, which was expressed in the index of adhesiveness of microorganisms (IAM) [28]. IAM is the average number of microbial cells attached to one erythrocyte involved in the adhesion process. This parameter was determined on 50 erythrocytes, manually counted through all glass slides.

A suspension of 1–2 daily *E. coli* at a concentration of 10^9^ cells/mL and a suspension of erythrocytes at a concentration of 100 million/mL were prepared in buffered saline (0.1 mL of phosphate Na in saline, pH = 7.2–7.3). To set up the experiment, 0.5 mL of erythrocytes, suspension of microorganisms, and LPS solutions were added to the tubes. After that, a smear was prepared, dried at room temperature, fixed and stained according to Romanovsky–Gimse.

### 2.15. Statistical Analysis

Statistical processing of the experimental data was performed using Student’s criterion (t-test). All results were presented as mean ± standard error of the mean (SEM). Differences with a value of *p* < 0.05 were considered to be statistically significant.

## 3. Results

### 3.1. Isolation and Chemical Composition of the LPS

The LPS was isolated from *P. agglomerans* 8488 cells by the phenol-water method with a yield of 10.9%. LPS preparation contained 38% carbohydrates, traces of proteins, and 3.4% nucleic acids after purification by precipitation with aqueous 50% trichloroacetic acid and ultracentrifugation. In a non-degraded LPS preparation, 2-keto-3-deoxyoctonic acid was found at 0.8%. The predominant LPS monosaccharides were mannose (30.9%), fucose (25.9%), rhamnose (24.7%) and glucose (12.8%) (Figure 1A). Significantly lower amounts of galactose (2.9%) and ribose (2.8%) were also detected. The predominant fatty acids of the *P. agglomerans* 8488 LPS were 3-hydroxytetradecanoic acid 14:0 (3–OH) (34.9%) and dodecanoic acid 12:0 (31.5%). In addition, 2-hydroxytetradecanoic acid 14:0 (2–OH) was detected at the concentration of 3.8% (Figure 1B). The fatty acid and monosaccharide compositions of the *P. agglomerans* strain 8488 LPS were similar to previously described for the *P. agglomerans* strain 7969 LPS [12].

### 3.2. Analysis of the Lipid A Structure

The lipid A structure is diverse among different genera, and in some cases within species of the same genus. This diversity is related to phosphate substitutions, as well as the number, type and distribution of fatty acids. The mass spectrum of lipid A after hydrolysis with 2% AcOH showed a peak at *m/z* 1796.21 (Figure 2). This peak was assigned to biphosphorylated hexaacylated species with the following fatty acid composition: four 14:0 (3-OH), one 14:0, and one 12:0 (calculated mass 1797.22 Da), which is typical for *E.coli* lipid A.

The spectrum contained main peaks corresponding to penta-acylated derivative *m/z* 1568.00 without one of the 14:0 (3–OH) residue and 14:0 replaced by 14:1 residue (calculated mass 1569.01 Da), tetra-acylated species lacking one 14:0 (3–OH) and 14:0 residues (calculated mass 1360.83 Da), and tri-acylated species with calculated mass of *m/z* 1133.62 lacking one more 14:0 (3–OH) residue (calculated mass 1134.63 Da). Lipid A triacyl species often accompany higher acylated variants.

### 3.3. Elucidation of O-Polysaccharide Structure

The LPS was isolated from bacterial cells by the phenol-water extraction and hydrolyzed under mild acid conditions. The OPS was isolated by gel-permeation chromatography (GPC). A sugar analysis of the OPS by gas-liquid chromatography (GLC) of the alditol acetates derived by full acid hydrolysis revealed the presence of rhamnose (Rha), mannose (Man), fucose (Fuc) and 2-acetamido-2-deoxy-glucose (GlcNAc) residues. The GLC of the acetylated (*S*)-2-octyl glycosides showed that mannose had the D configuration. The absolute configuration of the other sugars was determined based on the glycosylation effects on the ^13^C-NMR chemical shifts [29] as summarized and calculated by the GODDESS (Glycan-Optimized Database-Driven Empirical Spectrum Simulation) NMR (Nuclear Magnetic Resonance) simulation service [30].

A purified OPS fraction was analyzed by 1D and 2D NMR spectroscopy. The ^1^H-NMR spectrum of the OPS showed four peaks in the anomeric region (δ 4.74–5.07) from which followed the number of sugar residues in the OPS repeated unit. Two peaks at δ 1.18-1.29 corresponded to H-6 of 6-deoxy sugars, one peak of –CH_3_ to an N-acetyl group (δ 2.02), and peaks at δ 3.47–4.35 were assigned to ring protons (Figure 3).

The ^13^C-NMR spectrum of the OPS contained peaks at δ 101.1–103.6 (four anomeric carbons), peaks of CO and –CH_3_ of an N-acetyl group at δ 176.1 and δ 23.6, respectively, –CH_3_ of deoxy sugars at δ 16.6–17.9, one peak of carbon linked to nitrogen at δ 56.8 and peaks of ring carbons at δ 56.9–81.6. the ^1^H and ^13^C-NMR spectra were assigned by 2D COSY (Correlation Spectroscopy, TOCSY (Total Correlation Spectroscopy) and HSQC (Heteronuclear Single-Quantum Correlation) experiments, and four sugar spin systems were found and assigned. (Table 2).

The anomeric configuration of the monosaccharides was established by the C-5 chemical shifts compared with published data of the corresponding α- and β-pyranosides [31,32] and averaged by the GODDESS NMR simulation service [33]. These α-configurations were confirmed by H-1/H-2 correlations in the ROSEY (Rotating-frame Overhauser Spectroscopy) spectrum and the β-linkage of GlcNAc*p* was corroborated by the presence of H-1/H-5 cross-peaks in the ROSEY spectrum and by a large *J*_1,2_ value (8 Hz). The ^13^C-NMR chemical shift data revealed the substitution patterns of the monosaccharides by low-field displacements of the signals for C-3 of Rha*p*, Fuc*p* and GlcNAc*p* and C-6 of Man*p*, as compared to their positions in the corresponding non-substituted monosaccharides [31,32,33].

The absence of signals in the characteristic region of furanoses indicated that all sugar residues were in the pyranose form [31,32]. The positions of glycosylation were confirmed, and the sequence of the monosaccharide residues was established by correlations between the anomeric protons and protons at the linkage carbons in the ROESY and HMBC spectrum (Table 3). The following cross-peaks were observed: A H-1/B H-6a,b, B H-1/C H-3, C H-1/D H-3, D H-1/A H-3 (Figure 3) in the ROESY spectrum.

Based on the data obtained, it was concluded that the O-polysaccharide of *P. agglomerans* 8488 (top) has the structure shown in Figure 4, and a similar structure to the O–polysaccharide of *P. agglomerans* 7969 (bottom).

### 3.4. Biological Properties of the LPS

*P. agglomerans* 8488 was sensitive to polymyxin B, which showed a large area (d = 44 mm) of bacterial growth delay. The sensitivity to polymyxin B correlates with the lack of any positively charged substituents at the phosphate groups of lipid A of LPS that was demonstrated by ESI MS analysis. Indeed, substituents of this type are known to prevent binding of polycationic antibiotics, such as polymyxins, to the bacterial cell surface and thus to provide the antibiotic resistance.

The native LPS of *P. agglomerans* 8488 showed (Table 4) a relatively low level of toxicity in mice (LD_50_ = 105 µg/mice or 5 mg/kg). However, it was almost twice as toxic as the LD_50_ for *P. agglomerans* 7969 LPS (LD_50_ = 250 µg/mice or 11.9 mg/kg). The comparison of the toxicity of the native and modified LPS indicated that the modification of LPS by six tin (IV) complexes and two germanium complexes tested, reduced its toxicity [14]. Moreover, the presence of different substituents (2-OH, 2-OH-5-Br, 2-OCH_3_, 2-NH_2_) in the hydrazone molecules altered the toxicity of the modified *P. agglomerans* 8488 LPS. In addition, the LPS modified by two complex compounds of tin and one complex compounds of germanium were more toxic than native ones [14].

Pyrogenicity assays revealed that the *P. agglomerans* 8488 LPS was less pyrogenic compared to pyrogenal (LPS of *Salmonella typhi*) and *P. agglomerans* 7969 LPS with a similar structure. The temperature difference in the 3-h observation period for the experimental group of animals did not exceed 0.5 °C (the limit of the physiological norm), which indicated that LPS was not pyrogenic (Figure 5). Moreover, LPS of *P. agglomerans* 8488 lowered the body temperature for the 1- and 3-h time points.

The ability of LPS to bind cells due to adhesion is important for the pathogenicity of Gram-negative bacteria. The LPS of *P. agglomerans* 8488 reduced the index of adhesiveness of the *E. coli* ATCC 25922 cells (Table 5). Less interaction occurred between the surface structures of erythrocytes and *E. coli* cells at higher LPS concentration in the mixture. The index of adhesiveness of microorganisms (IAM) under the influence of the *P. agglomerans* LPS at the concentration of 3 mg/mL was 2.12, twofold less when compared to the control. In addition, the decrease in IAM for *P. agglomerans* 8488 was lower than *P. agglomerans* 7969 LPS (IAM=1.6 [12]), despite their similar LPS structures. It is possible that subtle differences in their structures can affect the adhesion processes.

O-antigen is the most variable portion of LPS and defines the serological specificity, which is used for bacterial serotyping. Ouchterlony double immunodiffusion in agar showed that the *P. agglomerans* 8488 LPS exhibited antigenic activity in the homologous system (Figure 6). As seen in Figure 6, O-antiserum against *P. agglomerans* 8488 cross-reacted with the LPSs of *P. agglomerans* 7969 and 8490. These data testified the presence of a common antigenic determinant(s) in the OPS of these LPS and indicated that these strains belong to the same serogroup. Modification by complex compounds of germanium and tin did not affect serological specificity [14], suggesting that the complex compounds most likely did not bind to OPS.

## 4. Discussion and Conclusions

Bacterial cell surface components that are in direct contact with the environment participate in many processes that determine the biological properties of microorganisms, the nature of the relationship between them, as well as micro- and macro-organisms in biocenosis. Among them, LPS is of particular interest. It consists of a hydrophobic part lipid A, which is embedded in the outer membrane of Gram-negative bacteria and hydrophilic core oligosaccharide and OPS (O-antigen), whose chains are exposed to the environment. There are many structural variants of the LPS due to the diversity of the chemical compositions of the polysaccharide moiety as well as significant differences in the fine structure of lipid A.

In the present study, the *P. agglomerans* 8488 LPS was isolated and purified and its chemical structure was elucidated. The predominant monosaccharides found in the LPS were mannose, fucose, rhamnose and glucosamine. The content of 2-keto-3-deoxyoctulosonic acid (KDO), a typical LPS component of Gram-negative bacteria, was low. A poor detection of KDO in the LPS might be due to its substitution at position 4 (or 5) by the phosphate group and at position 5 (or 7) by a carbohydrate chain, making impossible the formation of its fragment, which gives a positive reaction with thiobarbituric acid in the color assay [19]. The predominant fatty acid in the LPS is 14:0 (3-OH), which is a signature of lipid A of the *Enterobacteriaceae* family.

Lipid A is the most conserved part of the LPS and represents bisphosphorylated disaccharide of D-glucosamine acylated with 3-hydroxy fatty acids at positions 2 and 3 of both GlcN residues by amide and ester linkages. Despite being the most structurally conserved LPS component, lipid A shows a considerable structural microheterogeneity, which depends on various factors, including bacterial adaptation to changing environmental conditions and incompleteness of biosynthesis [34]. ESI MS analysis showed that lipid A of *P. agglomerans* 8488 was represented by hexa-, penta-, tetra- and tri-acylated species. Previously, we found that lipid A’s of four other strains of *P. agglomerans* were represented by hexa-, penta-, and tetra-acylated species [12,13,35]. The tri-acylated form of lipid A was detected only in the strain studied in this work.

The phosphate groups in lipid A can be substituted by polar or other groups, which change biological properties of not only LPS molecule but also the whole bacterial cell. For instance, a substituent (4-amino-4-deoxy-L-arabinose) on glucosamine II 4’-phosphate is responsible for bacterial resistance to some polycationic antibiotics, such as polymyxin B [34,36]. Our results on the sensitivity of *P. agglomerans* 8488 to polymyxin B are in agreement with the data of negative-ion mode ESI-MS analysis of lipid A, in which no 4-amino-4-deoxy-L-arabinose at the phosphate group was found.

The endotoxic properties of the LPS depend on the degree of the lipid A acylation [37]. A comparative study of the lipid A structure of six studied *P. agglomerans* strains suggests that the LPS toxicity and pyrogenicity depend not only on the degree of lipid A acylation, but also on the qualitative composition of fatty acids. Thus, lipid A of two *P. agglomerans* strains, P324 and 8488, contains 18:0 and 14:1 residues, respectively, which are absent from lipid A of other strains studied that were demonstrated to be most toxic [12,13,35]. Further studies of other *P. agglomerans* strains are needed for final conclusions to be made.

Differences in the toxicity and pyrogenicity of the LPSs of *P. agglomerans* 7969 and 8488 may be associated with differences in their lipid A structures. Thus, lipid A of *P. agglomerans* 7969 is represented only by hexa- and tetra-acylated species, whereas lipid A of *P. agglomerans* 8488 shows a higher degree of heterogeneity being represented by hexa-, penta-, tetra- and tri-acylated species.

One of the ways to change the functional and biological properties of lipopolysaccharides is chemical modification. The modifiers used were complexes of germanium and tin. The findings suggest that some complexes were sufficiently effective modulators of LPS toxicity. This is probably due to the properties of complexes of bioactive metals with hydrazones, which have antimicrobial, antitumor, anti-inflammatory activities due to the presence in their structure of an analog of the peptide group C(O)NH [38]. It should be noted that the different effects of the complexes that were used for LPS modifications may be caused not only by structural features of complex compounds, but also by lipids A of LPS of various P. agglomerans strains.

The OPS structure of *P. agglomerans* 8488 almost completely coincides with that of *P. agglomerans* 7969 [12]. The differences are in the non-stoichiometric substitution of the GlcNAc residue with glycerol phosphate at position 6 in the latter and the absolute configuration of the Fuc residue. The reevaluation of the NMR spectra of the OPS of *P. agglomerans* 7969 using the GODDESS NMR simulation service [30] showed that in this OPS, as in the OPS of *P. agglomerans* 8488, Fuc has the L-configuration rather than the D-configuration, as described earlier [12].

Subtle variations in OPS structures are the basis for the development of intraspecies serological classification schemes for Gram-negative bacteria. The first serological classification scheme based on the structures of OPS was proposed by Kauffmann and White for the *Salmonella* genus [39]. The authors showed that the serotypes of *Salmonella* genus members are determined by the presence of substituents at mannose or galactose. Significant progress made in the last decades in studying OPS structures has allowed the determining of 46 serogroups for *Salmonella enterica*, 46 for *Shigella* representatives, 76 for *Proteus*, 61 for *Providencia* and 39 *Hafnia*. A larger-scale study identified more than 180 serogroups for *E. coli* [40].

O-antiserum against *P. agglomerans* 8488 exerted a serological cross-reactivity with the LPS of two other *P. agglomerans* strains studied (7969 and 8490). Therefore, the three strains could be assigned to the same serogroup. Differences in the OPS structures of these strains, such as the substitution of GlcNAc residue by glycerol phosphate in strains 7969 and 8490, had no effect on the formation of antigen–antibody complexes in cross-reactions, suggesting that glycerol phosphate is no part of any antigenic determinant.

A serological study of the LPSs from 14 strains of *P. agglomerans* showed that only some of them exerted cross-reactivity. These findings allowed division, for the first time, of the 14 *P. agglomerans* strains into 10 serogroups based on the LPS O-antigenicity. Thus, further studies are necessary to reveal the cross-reactive epitope(s) within the corresponding OPSs and to estimate the level of the immunochemical heterogeneity of *P. agglomerans* strains.

The data found in literature and our own experimental results indicates the importance of the study of *P. agglomerans* LPS. Our results on the ability of the modified LPS which have lost their toxicity to block the toxic effects could stimulate the development on their basis of new drugs with various applications. In conclusion, the discovery of the heterogeneity of LPS, distinguished by unique structures of O-specific polysaccharides, lipids A, characterized by different degrees of acylation, as well as serological activity and endotoxic properties, make an essential contribution to the multidisciplinary characterization of the *P. agglomerans* species.

## Figures and Tables

**Figure 1 biomolecules-10-00804-f001:**
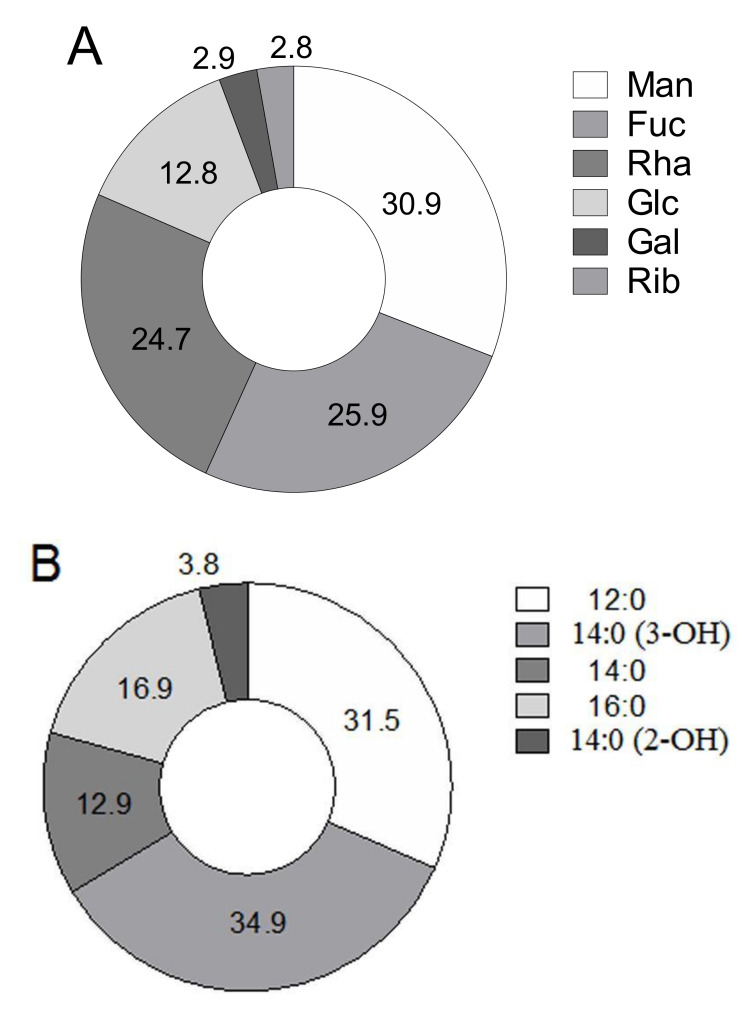
The monosaccharide (**A**) and fatty acid (**B**) compositions of *P. agglomerans* 8488 LPS. The monosaccharide and fatty acid compositions of LPS were expressed in percentage to the total sum of peak areas. Man = mannose, Fuc = fucose, Rha = rhamnose, Glc = glucose, Gal = galactose, Rib = ribose; 14:0 (3–OH)-3-hydroxytetradecanoic acid, 12:0—dodecanoic acid, 14:0—tetradecanoic acid, 16:0—hexadecanoic acid, 14:0 (2–OH)-2-hydroxytetradecanoic acid.

**Figure 2 biomolecules-10-00804-f002:**
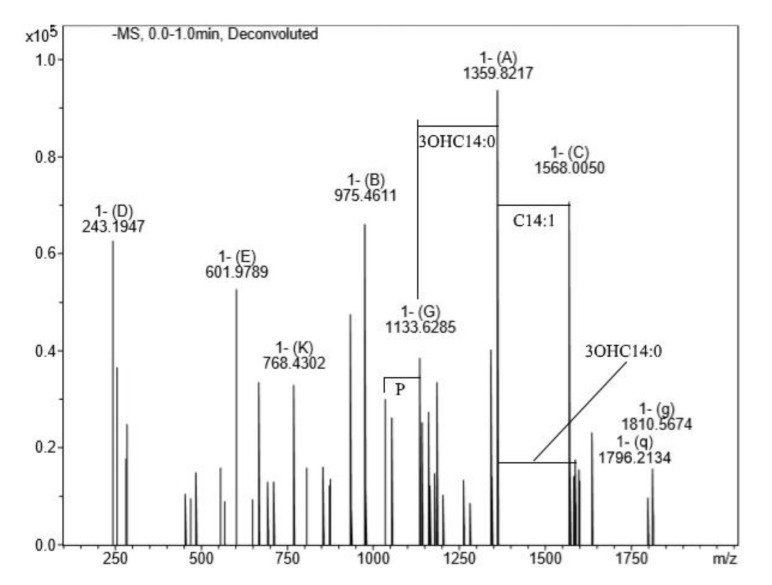
The mass spectrum of *P. agglomerans* 8488 lipid A. The major peak at *m/z* 1796.21 was assigned to the biphosphorylated hexaacylated species with fatty acid composition as follows: four 14:0 (3–OH), one 14:0, and one 12:0. The main peaks at *m/z* 1568.00 were assigned to penta-acylated derivative without one of the 14:0 (3–OH) and 14:0 replaced by 14:1, tetra-acylated species lacking one 14:0 (3–OH) and 14:0, and tri-acylated species at 62 *m/z* 1133 lacking one more 14:0 (3–OH).

**Figure 3 biomolecules-10-00804-f003:**
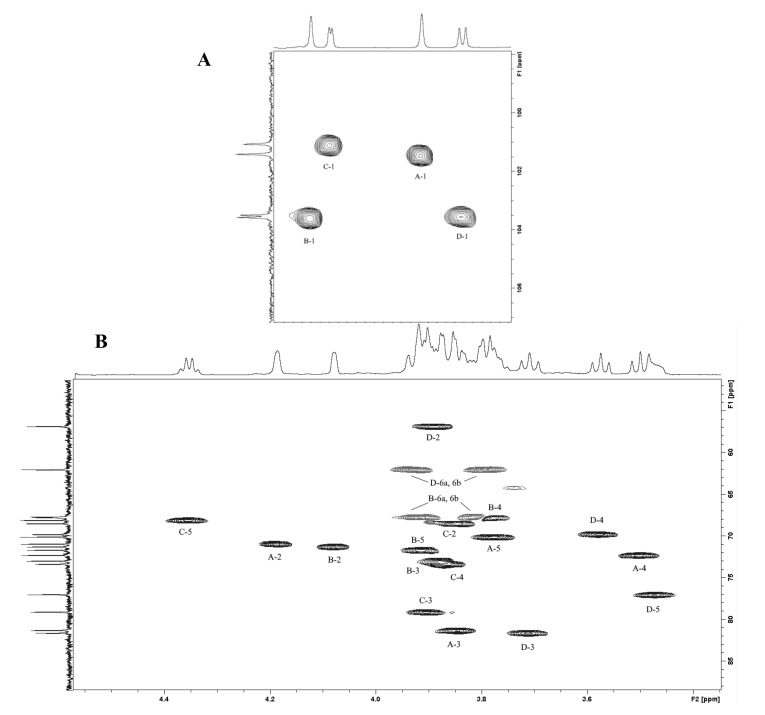
Part of a 2D ^1^H,^13^C HSQC spectrum of *P. agglomerans* 8488 OPS. The corresponding parts of the ^13^C and ^1^H-NMR spectra are displayed along the vertical F1 and horizontal F2 axes, respectively. Arabic numerals refer to the H/C pairs in the OPS components denoted as indicated in Table 1. (**A**) region of the anomeric atom signals, (**B**) other sugar ring atom signals region.

**Figure 4 biomolecules-10-00804-f004:**
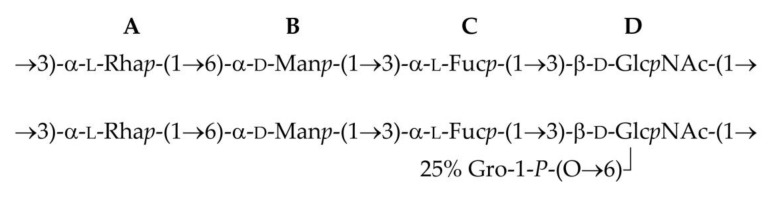
Structure of the O-polysaccharide of *P. agglomerans* 8488 (top) and revised structure of the O-polysaccharide of *P. agglomerans* 7969 (bottom).

**Figure 5 biomolecules-10-00804-f005:**
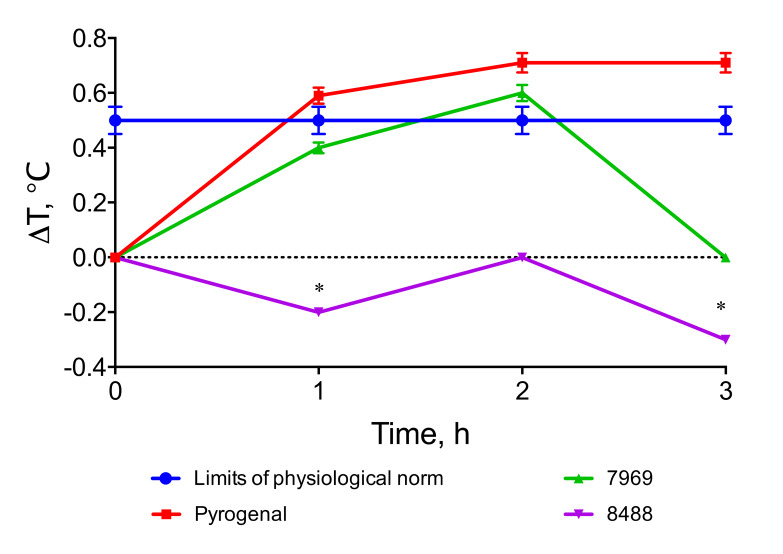
Pyrogenic effect of *P. agglomerans* 8488 and 7969 LPSs. The time of the temperature measurements is indicated on the X-axis, and the temperature change between measurements on the Y-axis. * a significant difference. Data are presented as means ± SEM.

**Figure 6 biomolecules-10-00804-f006:**
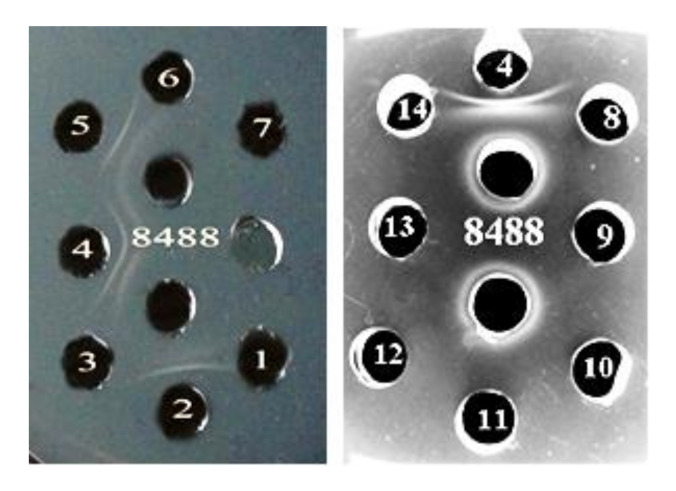
The reaction of Ouchterlony double immunodiffusion in agar with O-antiserum to *P. agglomerans* 8488 (two central holes) and LPS from *P. agglomerans* 7960a (1), 7969 (2), 8456 (3), 8488 (4), 8490 (5), 8606 (6), 8674 (7), P1a (8), P324 (9), 7406 (10), 7604 (11), 9637 (12), 9649 (13), 9668(14). The white bars between the holes indicate the presence of an antigen-antibody complexes.

**Table 1 biomolecules-10-00804-t001:** Complexes of tin (IV) and germanium (IV).

No.	Formula of the Complex	Scheme of Complexes Structure	No.	Formula of the Complex	Scheme of Complexes Structure
1	[SnCl_4_(Bdb·H)]	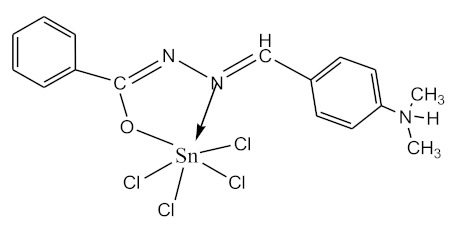	11	[SnCl_3_(HBs)]	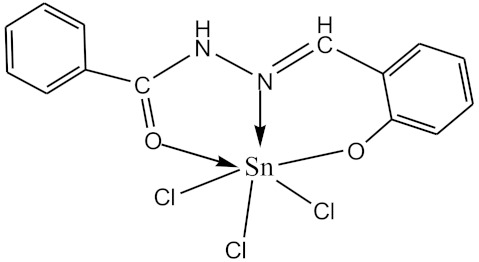
2	[SnCl_4_(2-OH-Bdb·H)]	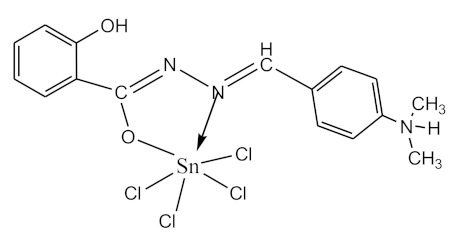	12	[SnCl_3_(2-OH-HBs)]	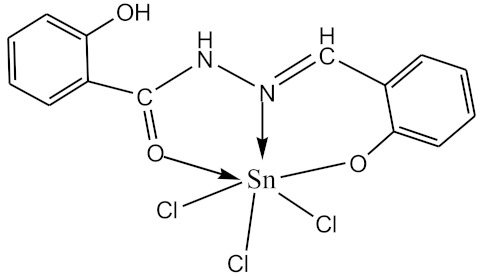
3	[SnCl_4_(2-OH-5-Br-Bdb·H)]	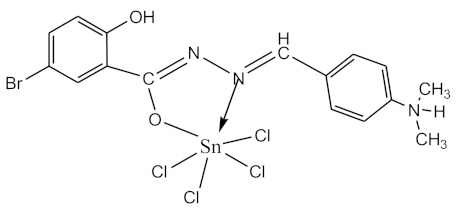	13	[SnCl_3_(2-OH-5-Br-HBs)]	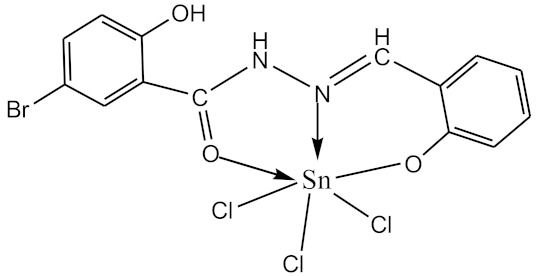
4	[SnCl_4_(2-OCH_3_-Bdb·H)]	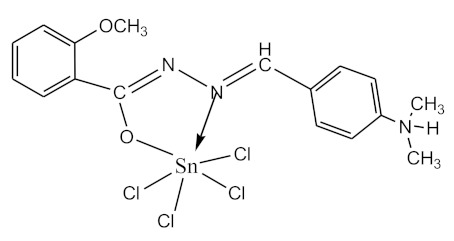	14	[SnCl_3_(2-OCH_3_-HBs)]	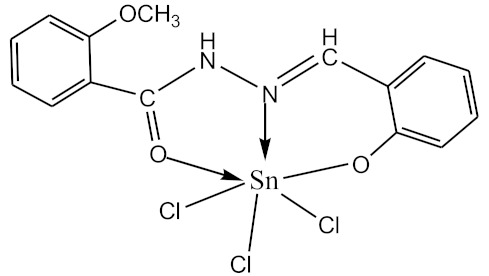
5	[SnCl_4_(2-NH_2_-Bdb·H)]	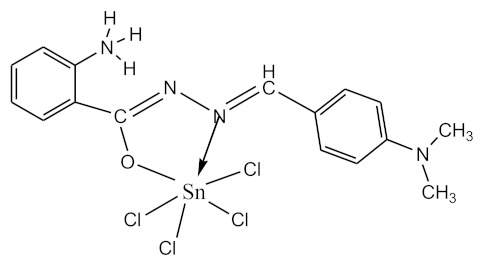	15	[SnCl_3_(2-NH_2_-Bs·H)]	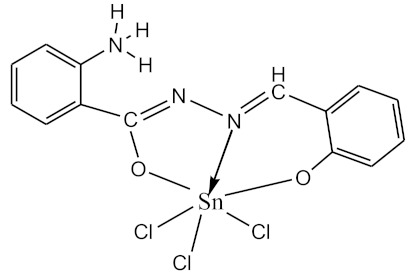
6	[SnCl_4_(Ndb·H)]	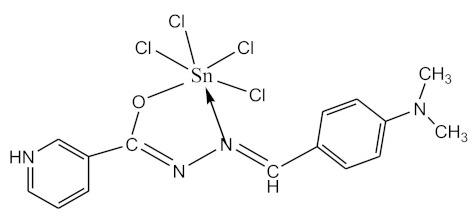	16	[SnCl_3_(Ns∙H)]	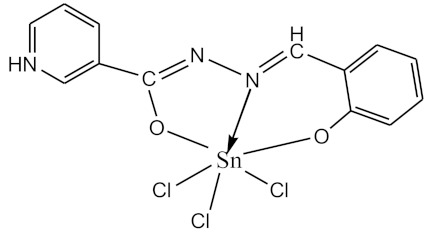
7	[SnCl_4_(Idb·H)]	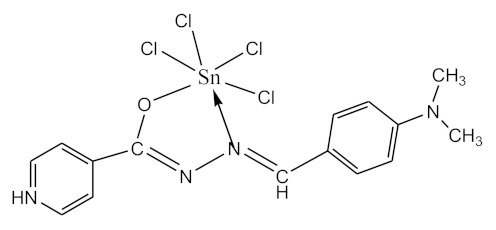	17	[SnCl_3_(Is·H)]	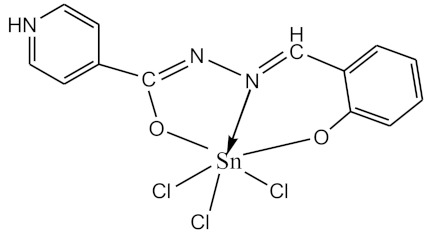
8	[SnCl_4_(Ldb·H)]	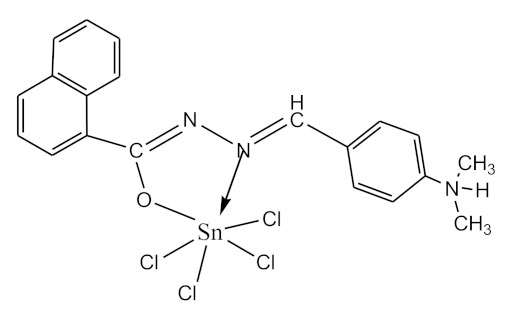	18	[SnCl_3_(HLs)]	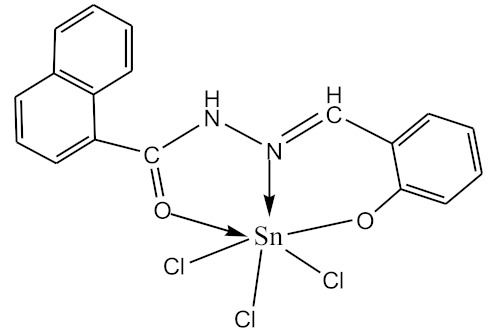
9	[Ge(2-OH-Bs)_2_]	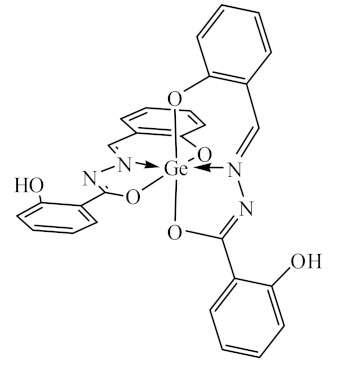	19	[Ge(Ns)_2_]	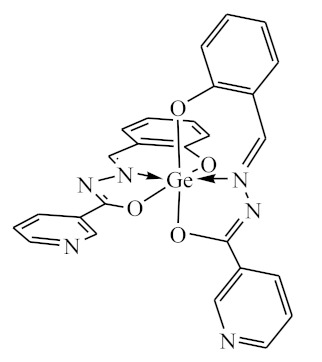
10	[Ge(2-NH_2_-Bs)_2_]	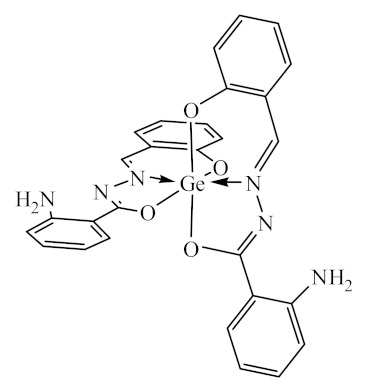	20	[Ge(Is)_2_]	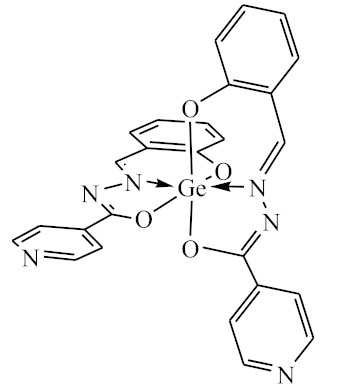

**Table 2 biomolecules-10-00804-t002:** The ^1^H and ^13^C-NMR chemical shifts (δ, ppm) of *P. agglomerans* 8488 O-polysaccharide.

Sugar Residue	Unit	*H-1* C-1	*H-2* C-2	*H-3* C-3	*H-4* C-4	*H-5* C-5	*H-6 (a,b)* C-6
→3)-α-L-Rha*p*-(1→	A	*4.84* 101.4	*4.19* 71.0	*3.84* 81.3	*3.50* 72.2	*3.77* 70.1	*1.29* 17.9
→6)-α-D-Man*p*-(1→	B	*5.07* 103.6	*4.08* 71.3	*3.87* 72.8	*3.78* 67.6	*3.92* 71.9	*3.92; 3.82* 67.6
→3)-α-L-Fuc*p*-(1→	C	*5.03* 101.1	*3.84* 67.4	*3.90* 78.9	*3.88* 73.0	*4.35* 67.0	*1.18* 16.6
→3)-β-D-GlcNAc*p*-(1→	D	*4.74* 103.6	*3.88* 56.8	*3.71* 81.5	*3.57* 69.8	*3.47* 77.0	*3.93; 3.79* 62.2

**Table 3 biomolecules-10-00804-t003:** Correlations for H-1 in the 2D ROESY spectrum of the O-polysaccharide of *P. agglomerans* 8488.

Anomeric Atom in Sugar Residue	Correlation(s) to Atoms in Sugar Residue(s) (δ)	
	ROESY	HMBC
**A** H-1 (*4.84*)	**A** H-2 (*4.19*), **B** H-6a (*3.92*), **B** H-6b (*3.82*),	**B** C-6 (67.6), **A** C-5 (70.1), **A** C-3 (81.3)
**A** C-1 (101.4)		**B** H-6a,b (*3.92*; *3.82*)
**B** H 1 (*5.07*)	**B** H-2 (*4.08*), **B** H-3 (*3.87*), **C** H-3 (*3.90*), **C** H-4 (*3.88*)	**C** C-3 (78.9), **C** C-4 (73.0), **B** C-5 (71.9)
**B** C-1 (103.6)		**C** H-3 (*3.90*), **C** H-4 (*3.88*), **C** H-2 (*3.84*)
**C** H 1 (*5.03*)	**C** H-3 (*3.90*), **C** H-4 (*3.88*), **D** H-3 (*3.71*), **D** H-4 (*3.57*)	**C** C-3 (78.9), **D** C-3 (81.5)
**C** C-1 (101.1)		**D** H-3 (*3.71*)
**D** H 1 (*4.74*)	**D** H-2 (*3.88*), **D** H-3 (*3.71*), **D** H-4 (*3.57*), **D** H-5 (*3.47*), **A** H-2 (*4.19*), **A** H-3 (*3.84*)	**A** C-3 (81.3)
**D** C-1 (103.6)		**A** H-3 (*3.84*), **D** H-2 (*3.88*)

Where: **A** = Rha, **B** = Man, **C** = Fuc, **D** = GlcNAc.

**Table 4 biomolecules-10-00804-t004:** Determination of toxicity of the *P. agglomerans* 8488 LPS.

LPS	The Number of Dead Animals (Numerator) and the Total Number of Animals (Denominator) after Injection of Different Doses (d, μg/mouse) of the Tested LPS				LD_50_	
	d = 62.5	d = 125	d = 250	d = 500	μg/mouse	mg/kg
*P. agglomerans* 8488	5/10	5/10	7/10	10/10	105	5

**Table 5 biomolecules-10-00804-t005:** Effect of different concentrations of *P. agglomerans* P324 LPS on the index of adhesiveness of *E. coli*.

Concentration of LPS in Solution, mg/mL	Index of Adhesiveness of Microorganism, Usnit
Control (without LPS)	4.89
0.012	4.39
0.023	4.3
0.047	4.1
0.094	3.45
0.188	2.68
0.375	2.53
0.75	2.19
1.5	2.13
3	2.12

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
