# Peer review of "Structure of O-Polysaccharide and Lipid A of Pantoea Agglomerans 8488"

_biomolecules, 2020, doi:10.3390/biom10050804_

Round 1

Reviewer 1 Report

The article “Structure of O-Polysaccharide and Lipid A of  Pantoea agglomerans 8488” by T.Butylina et al. describes the complete chemical characterization and numerous biological activities (toxicity, pyrogenicity, effect on bacterial adhesion to erythrocytes) of a lipopolysaccharide (LPS) from a strain of Pantoea agglomerans, isolated from oat. The effect of LPS modifications by synthetic complexes of Ge (IV) and Sn (IV) on toxicity in mice was studied. Cross-reactivity of O-antiserum to P.  agglomerans 8488 with LPS from other P.  agglomerans strains was tested and matched with data on chemical structures of OPS, which is important for serological classification of P. agglomerans.

This is a complete, innovative and multidisciplinary study. Experiments are well performed and the presentation of results is clear.

However, certain points should be corrected or developed, prior to publications.

General comments

  • Chemical detoxification of LPS with synthetic metal-containing complexes seems to be an interesting and innovative approach. In my opinion, it would be useful to give some background and a literature overview of this methodology, and give more attention to the corresponding results.
  • Figure 3 is missing in my copy of the manuscript

Specific comments

Line 31: It might be useful to give the former name/names for Pantoea agglomerans (Enterobacter agglomerans, Erwinia herbicola) [ ex. Gavini et al., International Journal of Systematic Bacteriology 39 (3), 337-345(1989)]

Lines 53-60 or Discussion section: Ref. [3] explores the use of oral administration of LPS from a of P. agglomerans strain for treatment of cancer patients, as activating immune cells and inducing foreign body exclusion, inflammation and tissue repair. I suggest that discussing this point would emphasize the value of the findings.

Line 185:  please check IAM –Index of Adhesion of Microorganisms?

Line 243 and Fig. 2 : “major peak at m/z 1796.21” is not seen in Fig. 2, please check.

Table 3. Please correct the editing of the table. Also, notes are not defined.

Line 292. Linkage determination: ROESY correlations shown in Tab. 3 are ambiguous for linkages B—C; C—D and D—A. I would suggest to put the 13C NMR chemical shifts data after the results of the ROESY experiments, as a clarification of the ambiguity. What are the results of HMBC experiments?

Fig. 4, lines 300-305 : Gro-1-P-(O→6) [11] instead of Cro-1-(O→6).  Labeling the sugar residues A, B, C, D in Fig. 4 would simplify the understanding of the text.

Lines 312-314

**The native LPS of P. agglomerans 8488 showed a relatively low level of toxicity in mice  (LD50=105 μg/mice or 5 mg/kg). However, it was almost twice as high as the LD50 for P. agglomerans 7969 LPS (LD50=250 μg/mice or 11.9 mg/kg).** Please verify this statement

Lines 314-319: Please add the results of the effect of metal complexes on LPS toxicity

Fig. 6. On the left photograph, it is difficult to distinguish the white bars. Hole 4, positive control?

Author Response

General comments

Chemical detoxification of LPS with synthetic metal-containing complexes seems to be an interesting and innovative approach. In my opinion, it would be useful to give some background and a literature overview of this methodology, and give more attention to the corresponding results.

In the "Discussion" section added:

One of the ways of changes in the functional and biological properties of lipopolysaccharides is the chemical modification. As modifiers were used complexes of germanium and tin. The findings suggest that some complexes were sufficiently effective modulators of LPS toxicity. This is probably due to the properties of complexes of bioactive metals with hydrazones, which have antimicrobial, antitumor, anti-inflammatory activity, due to the presence in their structure of an analog of the peptide group C(O)NH [Rollas, S.; Guniz Kucukguzel, S. Biological Activities of Hydrazone Derivatives. Molecules 2007, 12, 1910–1939.]. It should be noted that the different effects of the complexes that were used for LPS modifications may be caused not only by structural features of complex compounds, but also by lipids A of LPS of various P. agglomerans strains.

 Figure 3 is missing in my copy of the manuscript. The Figure 3 was inserted into the text.

Specific comments

Line 31: It might be useful to give the former name/names for Pantoea agglomerans(Enterobacter agglomeransErwinia herbicola) [ ex. Gavini et al., International Journal of Systematic Bacteriology 39 (3), 337-345(1989)]

Agree. Added the phrase (Line 31) and a link to the reference list.

Lines 53-60 or Discussion section: Ref. [3] explores the use of oral administration of LPS from a of P. agglomerans strain for treatment of cancer patients, as activating immune cells and inducing foreign body exclusion, inflammation and tissue repair. I suggest that discussing this point would emphasize the value of the findings.

Thank you very much for the comment. It is difficult for us to compare our results with the results described in the article (Morishima, A.; Inagawa, H. Clinical effects of orally administered lipopolysaccharide derived from Pantoea agglomerans on malignant tumors. Anticancer Res 2016, 36, 3747-3752.). Since we did not carry out immunological studies and cannot claim that P. agglomerans 8488 LPS will exhibit the same effect as described in the article. In addition, the article does not have the LPS characterization and a description of its structure. Even in terms of structure, we cannot compare these LPS.

We have a written article “Anti-lipopolysaccharide antibodies and osmotic resistance of erythrocytes in healthy individuals and patients with B-non-hodgkin's lymphoma with different blood groups”, which is now at the printing stage in another journal. The aim of this work was to evaluate the presence of antibodies to LPS isolated from three strains of Enterobacteriaceae, representatives of Escherichia coli and Pantoea agglomerans 8488, from donors with different blood groups, and from patients with B-non-Hodgkin's lymphoma. And according to the results described in this article, we will make the following conclusions: “Blood group B (III) donors are characterized by a higher titer of anti-LPS antibodies to E. coli L-19 (LPS 1), which may indicate a higher sensitivity to LPS, although their blood donor levels have been increased compared to patients with B-NHL. This fact may need to be taken into account in blood transfusions in leukemia’s and to determine anti-LPS antibodies from donors.”

Line 185:  please check IAM –Index of Adhesion of Microorganisms?

IAM - Index of Adhesiveness of Microorganisms. Corrected.

Line 243 and Fig. 2 : “major peak at m/z 1796.21” is not seen in Fig. 2, please check.

The peak at m/z 1796.21 was not signed. Corrected.

Table 3. Please correct the editing of the table. Also, notes are not defined.

The Table 3 was corrected, data on HMBC experiment and notes were added

Line 292. Linkage determination: ROESY correlations shown in Tab. 3 are ambiguous for linkages B—C; C—D and D—A. I would suggest to put the 13C NMR chemical shifts data after the results of the ROESY experiments, as a clarification of the ambiguity. What are the results of HMBC experiments?

Data on the 13C NMR chemical shifts revealed positions of the substitution of the monosaccharides are indicated on the lines 336-339. Data of HMBC experiment are added to the Table 3 and to the text.

Fig. 4, lines 300-305 : Gro-1-P-(O→6) [11] instead of Cro-1-(O→6).  Labeling the sugar residues A, B, C, D in Fig. 4 would simplify the understanding of the text.

Corrected

Lines 312-314

**The native LPS of P. agglomerans 8488 showed a relatively low level of toxicity in mice  (LD50=105 μg/mice or 5 mg/kg). However, it was almost twice as high as the LD50 for P. agglomerans 7969 LPS (LD50=250 μg/mice or 11.9 mg/kg).** Please verify this statement

The LPS of P. agglomerans 8488 showed a relatively low level of the toxic activity (LD50=105 μg/mice or 5 mg/kg) whereas LD50 of other representatives of the Enterobacteriaceae family varies from 3.6 to 75 μg/mouse (Shubchinskiy, V. V. (2008). Lipopolysaccharide of Pragia fontium: Chemical characterization and biological activity, PhD Thesis, Kiev (in Ukrainian)). However, it was almost twice as toxic as P. agglomerans 7969 LPS. (LD50=250 μg/mice or 11.9 mg/kg). The lower the concentration of LPS is injected to mice, it is considered the more toxic (inverse proportionality). Corrected.

Lines 314-319: Please add the results of the effect of metal complexes on LPS toxicity

These data are presented in our article: Ref. 14. Bulyhina, T.V.; Varbanets, L.D.; Seyfullina, I.I.; Shmatkova, N.V. Functional and biological activity of Pantoea agglomerans lipopolysaccharides. Microbiol. Z. 2016, 78, 13-25.

Fig. 6. On the left photograph, it is difficult to distinguish the white bars. Hole 4, positive control?

Changed the brightness and contrast of the picture. The P. agglomerans 8488 LPS exhibited antigenic activity in the homologous system (Hole 4, positive control).

Reviewer 2 Report

LPS was isolated from P. аgglomerans 8488, characterized and analyzed the functional and biological activities. The peculiarity of OPS and lipid A structure was also determined by authors, which is a very interesting article. However, some questions need to resolve before publishing:

  1. Line 107, 121: min-1 or /min? Please normalize them in full text.
  2. There is no Figure 3, please provide it in the text.
  3. Line 298-299: please rewrite the sentence.
  4. Line 307: what is d=44 mm? please rewrite the sentence.
  5. Please give the figure of the survival of mice challenged with LPS if it is possible.
  6. Figure 6: please uniformly adjust the background color of left and right one if it is possible.
  7. Please give a conclusion in the end of text.
  8. Line 443, 457, 476, 481, 492, etc.: Please list the uniform format of references, such as full or abbreviated names of journals.
  9. Other minors:

Line 18: full name of NMR

Line 63: …degradation of the LPS

Line 97: P. agglomerans

Line 199, 327: LPS

Line 40, 56, 330, 359, 366, 406: Gram-

Author Response

  1. Line 107, 121: min-1 or /min? Please normalize them in full text.

Thanks for the adjustments. Normalized units of measure throughout the full text.

2. There is no Figure 3, please provide it in the text.

The Figure 3 was inserted into the text.

3.Line 298-299: please rewrite the sentence.

The sentence has been completed.

 4. Line 307: what is d=44 mm? please rewrite the sentence.

d=44 mm - The diameter of the zone of growth inhibition of bacterial cultures around the disk with the antibiotic. Indicated in the “Materials and Methods” section (line 153).

 5. Please give the figure of the survival of mice challenged with LPS if it is possible.

Added Table 4 (Determination of toxicity of the P. agglomerans 8488 LPS) that showed the survival rate of mice.

 6. Figure 6: please uniformly adjust the background color of left and right one if it is possible.

Changed the brightness and contrast of the picture.

 7. Please give a conclusion in the end of text.

Added conclusion in the end of text:

The data found in literature and our own experimental results indicate the importance of the study of P. agglomerans LPS. Our results on the ability of the modified LPS which have lost its toxicity to block the toxic effects could stimulate the development on their base of new drugs of various applications. In conclusion, discovery of the heterogeneity of LPS, distinguished by unique structures of O-specific polysaccharides, lipids A, characterized by different degrees of acylation, as well as serological activity and endotoxic properties make an essential contribution to the multidisciplinary characterization of the P. agglomerans species.

 8. Line 443, 457, 476, 481, 492, etc.: Please list the uniform format of references, such as full or abbreviated names of journals.

Checked the design of the reference list, corrected errors in the names of journals (Line 556, 560, 4587, 590, 593, 603, 608, 610, 612, 613, 616, 620, 623, 626, 648, 657, 663, 665-667, 672-674, 687, 786).

9. Other minors:

Line 18: full name of NMR – Corrected: NMR - nuclear magnetic resonance.

Line 63: …degradation of the LPS – Replaced “degradation of the lipopolysaccharide” by “degradation of the LPS”.

Line 97: P. agglomerans - Replaced “Pantoea. agglomerans” by “P. agglomerans”.

Line 199, 327: LPS - Replaced “lipopolysaccharide” by “LPS

Line 40, 56, 330, 359, 366, 406: Gram-. Corrected